# LEARNING LESS-OVERLAPPING REPRESENTATIONS

## ABSTRACT

In representation learning (RL), how to make the learned representations easy to interpret and less overfitted to training data are two important but challenging issues. To address these problems, we study a new type of regularization approach that encourages the supports of weight vectors in RL models to have small overlap, by simultaneously promoting near-orthogonality among vectors and sparsity of each vector. We apply the proposed regularizer to two models: neural networks (NNs) and sparse coding (SC), and develop an efficient ADMM-based algorithm for regularized SC. Experiments on various datasets demonstrate that weight vectors learned under our regularizer are more interpretable and have better generalization performance.

## 1 INTRODUCTION

In representation learning (RL), two critical issues need to be considered. First, how to make the learned representations more interpretable? Interpretability is a must in many applications. For instance, in a clinical setting, when applying deep learning (DL) and machine learning (ML) models to learn representations for patients and use the representations to assist clinical decision-making, we need to explain the representations to physicians such that the decision-making process is transparent, rather than being black-box. Second, how to avoid overfitting? It is often the case that the learned representations yield good performance on the training data, but perform less well on the testing data. How to improve the generalization performance on previously unseen data is important.

In this paper, we make an attempt towards addressing these two issues, via a unified approach. DL/ML models designed for representation learning are typically parameterized with a collection of weight vectors, each aiming at capturing a certain latent feature. For example, neural networks are equipped with multiple layers of hidden units where each unit is parameterized by a weight vector. In another representation learning model – sparse coding (Olshausen & Field, 1997), a dictionary of basis vectors are utilized to reconstruct the data. In the interpretation of RL models, a major part is to interpret the learned weight vectors. Typically, elements of a weight vector have one-to-one correspondence with observed features and a weight vector is oftentimes interpreted by examining the top observed-features that correspond to the largest weights in this vector. For instance, when applying SC to reconstruct documents that are represented with bag-of-words feature vectors, each dimension of a basis vector corresponds to one word in the vocabulary. To visualize/interpret a basis vector, one can inspect the words corresponding to the large values in this vector. To achieve better interpretability, various constraints have been imposed on the weight vectors. Some notable ones are: (1) Sparsity (Tibshirani, 1996) – which encourages most weights to be zero. Observed features that have zeros weights are considered to be irrelevant and one can focus on interpreting a few non-zero weights. (2) Diversity (Wang et al., 2015) – which encourages different weight vectors to be mutually "different" (e.g., having larger angles (Xie et al., 2015)). By doing this, the redundancy among weight vectors is reduced and cognitively one can map each weight vector to a physical concept in a more unambiguous way. (3) Non-negativeness (Lee & Seung, 1999) – which encourages the weights to be nonnegative since in certain scenarios (e.g., bag of words representation of documents), it is difficult to make sense of negative weights. In this paper, we propose a new perspective of interpretability: *less-overlapness*, which encourages the weight vectors to have small overlap in supports[1]. By doing this, each weight vector is anchored on a unique subset of observed features without being redundant with other vectors, which greatly facilitates interpretation. For example, if topic models (Blei et al., 2003) are learned in such a way, each topic

---

[1]The support of a vector is the set of indices of nonzero entries in this vector.

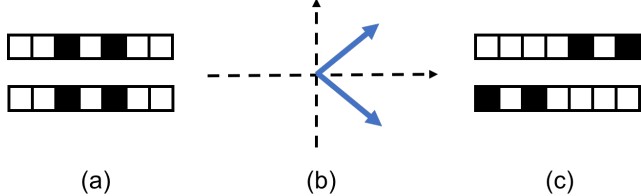

Figure 1: (a) Under L1 regularization, the vectors are sparse, but their supports are overlapped; (b) Under LDD regularization, the vectors are orthogonal, but their supports are overlapped; (c) Under LDD-L1 regularization, the vectors are sparse and mutually orthogonal and their supports are not overlapped.

will be characterized by a few representative words and the representative words of different topics are different. Such topics are more amenable for interpretation. Besides improving interpretability, less-overlapness helps alleviate overfitting. It imposes a structural constraint over the weight vectors, thus can effectively shrink the complexity of the function class induced by the RL models and improve the generalization performance on unseen data.

To encourage less-overlapness, we propose a regularizer that simultaneously encourages different weight vectors to be close to being orthogonal and each vector to be sparse, which jointly encourage vectors' supports to have small overlap. The major contributions of this work include:

- We propose a new type of regularization approach which encourages less-overlapness, for the sake of improving interpretability and reducing overfitting.
- We apply the proposed regularizer to two models: neural networks and sparse coding (SC), and derive an efficient ADMM-based algorithm for the regularized SC problem.
- In experiments, we demonstrate the empirical effectiveness of this regularizer.

## 2 METHODS

In this section, we propose a nonoverlapness-promoting regularizer and apply it to two models.

### 2.1 NONOVERLAPNESS-PROMOTING REGULARIZATION

We assume the model is parameterized by $m$ vectors $\mathcal{W} = \{\mathbf{w}_i\}_{i=1}^m$. For a vector $\mathbf{w}$, its *support* $s(\mathbf{w})$ is defined as $\{i|w_i \neq 0\}$ – the indices of nonzero entries in $\mathbf{w}$. We first define a score $\tilde{o}(\mathbf{w}_i, \mathbf{w}_j)$ to measure the overlap between two vectors:

$$\tilde{o}(\mathbf{w}_i, \mathbf{w}_j) = \frac{|s(\mathbf{w}_i) \cap s(\mathbf{w}_j)|}{|s(\mathbf{w}_i) \cup s(\mathbf{w}_j)|}. \tag{1}$$

which is the Jaccard index of their supports. The smaller $\tilde{o}(\mathbf{w}_i, \mathbf{w}_j)$ is, the less overlapped the two vectors are. For $m$ vector, the overlap score is defined as the sum of pairwise scores

$$o(\mathcal{W}) = \frac{1}{m(m-1)} \sum_{\substack{i \neq j}}^m \tilde{o}(\mathbf{w}_i, \mathbf{w}_j). \tag{2}$$

This score function is not smooth, which will result in great difficulty for optimization if used as a regularizer. Instead, we propose a smooth function that is motivated from $\tilde{o}(\mathbf{w}_i, \mathbf{w}_j)$ and can achieve a similar effect as $o(\mathcal{W})$. The basic idea is: to encourage small overlap, we can encourage (1) each vector has a small number of non-zero entries and (2) the intersection of supports among vectors is small. To realize (1), we use an L1 regularizer to encourage the vectors to be sparse. To realize (2), we encourage the vectors to be close to being orthogonal. For two sparse vectors, if they are close to orthogonal, then their supports are landed on different positions. As a result, the intersection of supports is small.

We follow the method proposed by Xie et al. (2017b) to promote orthogonality. To encourage two vectors $\mathbf{w}_i$ and $\mathbf{w}_j$ to be close to being orthogonal, one can make their $\ell_2$ norm $\|\mathbf{w}_i\|_2$, $\|\mathbf{w}_j\|_2$ close

to one and their inner product $\mathbf{w}_i^\top \mathbf{w}_j$ close to zero. Based on this, one can promote orthogonality among a set of vectors by encouraging the Gram matrix $\mathbf{G}$ ($G_{ij} = \mathbf{w}_i^\top \mathbf{w}_j$) of these vectors to be close to an identity matrix $\mathbf{I}$. Since $\mathbf{w}_i^\top \mathbf{w}_j$ and zero are off the diagonal of $\mathbf{G}$ and $\mathbf{I}$ respectively, and $\|\mathbf{w}_i\|_2^2$ and one are on the diagonal of $\mathbf{G}$ and $\mathbf{I}$ respectively, encouraging $\mathbf{G}$ close to $\mathbf{I}$ essentially makes $\mathbf{w}_i^\top \mathbf{w}_j$ close to zero and $\|\mathbf{w}_i\|_2$ close to one. As a result, $\mathbf{w}_i$ and $\mathbf{w}_j$ are encouraged to be close to being orthogonal. In (Xie et al., 2017b), one way proposed to measure the "closeness" between two matrices is to use the log-determinant divergence (LDD) (Kulis et al., 2009). The LDD between two $m \times m$ positive definite matrices $\mathbf{X}$ and $\mathbf{Y}$ is defined as $D(\mathbf{X}, \mathbf{Y}) = \text{tr}(\mathbf{X}\mathbf{Y}^{-1}) - \log\det(\mathbf{X}\mathbf{Y}^{-1}) - m$ where $\text{tr}(\cdot)$ denotes matrix trace. The closeness between $\mathbf{G}$ and $\mathbf{I}$ can be achieved by encouraging their LDD $D(\mathbf{G}, \mathbf{I}) = \text{tr}(\mathbf{G}) - \log\det(\mathbf{G}) - m$ to be small.

Combining the orthogonality-promoting LDD regularizer with the sparsity-promoting L1 regularizer together, we obtain the following LDD-L1 regularizer

$$\Omega(\mathcal{W}) = \text{tr}(\mathbf{G}) - \log\det(\mathbf{G}) + \gamma \sum_{i=1}^{m} |\mathbf{w}_i|_1 \qquad (3)$$

where $\gamma$ is a tradeoff parameter between these two regularizers. As verified in experiments, this regularizer can effectively promote non-overlapness. The formal analysis of the relationship between Eq.(3) and Eq.(2) will be left for future study. It is worth noting that either L1 or LDD alone is not sufficient to reduce overlap. As illustrated in Figure 1(a) where only L1 is applied, though the two vectors are sparse, their supports are completely overlapped. In Figure 1(b) where the LDD regularizer is applied, though the two vectors are very close to orthogonal, their supports are completely overlapped since they are dense. In Figure 1(c) where the LDD-L1 regularizer is used, the two vectors are sparse and are close to being orthogonal. As a result, their supports are not overlapped.

## 2.2 CASE STUDIES

In this section, we apply the LDD-L1 regularizer to two models.

**Neural Networks** In a neural network (NN) with $L$ hidden layers, each hidden layer $l$ is equipped with $m^{(l)}$ units and each unit $i$ is connected with all units in layer $l - 1$. Hidden unit $i$ at layer $l$ is parameterized by a weight vector $\mathbf{w}_i^{(l)}$. These hidden units aim at capturing latent features underlying data. For $m^{(l)}$ weight vectors $\mathcal{W}^{(l)} = \{\mathbf{w}_i^{(l)}\}_{i=1}^{m^{(l)}}$ in each layer $l$, we apply the LDD-L1 regularizer to encourage them to have small overlap. An LDD-L1 regularized NN problem (LDD-L1-NN) can be defined in the following way:

$$\min_{\{\mathcal{W}^{(l)}\}_{l=1}^L} \quad \mathcal{L}(\{\mathcal{W}^{(l)}\}_{l=1}^L) + \lambda \sum_{l=1}^L \Omega(\mathcal{W}^{(l)})$$

where $\mathcal{L}(\{\mathcal{W}^{(l)}\}_{l=1}^L)$ is the objective function of this NN.

**Sparse Coding** Given $n$ data samples $\mathbf{X} \in \mathbb{R}^{d \times n}$ where $d$ is the feature dimension, we aim to use a dictionary of basis vectors $\mathbf{W} \in \mathbb{R}^{d \times m}$ to reconstruct $\mathbf{X}$, where $m$ is the number of basis vectors. Each data sample $\mathbf{x}$ is reconstructed by taking a sparse linear combination of the basis vectors $\mathbf{x} \approx \sum_{j=1}^m \alpha_j \mathbf{w}_j$, where $\{\alpha_j\}_{j=1}^m$ are the linear coefficients and most of them are zero. The reconstruction error is measured using the squared L2 norm $\|\mathbf{x} - \sum_{j=1}^m \alpha_j \mathbf{w}_j\|_2^2$. To achieve sparsity among the codes, L1 regularization is utilized: $\sum_{j=1}^m |\alpha_j|_1$. To avoid the degenerated case where most coefficients are zero and the basis vectors are of large magnitude, L2 regularization is applied to the basis vectors: $\|\mathbf{w}_j\|_2^2$. We apply the LDD-L1 regularizer to encourage the supports of basis vectors to have small overlap. Putting these pieces together, we obtain the LDD-L1 regularized SC (LDD-L1-SC) problem

$$\min_{\mathbf{W}, \mathbf{A}} \quad \frac{1}{2}\|\mathbf{X} - \mathbf{W}\mathbf{A}\|_F^2 + \lambda_1 |\mathbf{A}|_1 + \frac{\lambda_2}{2}\|\mathbf{W}\|_F^2 + \frac{\lambda_3}{2}(\text{tr}(\mathbf{W}^\top \mathbf{W}) - \log\det(\mathbf{W}^\top \mathbf{W})) + \lambda_4 |\mathbf{W}|_1$$
$$(4)$$

where $\mathbf{A} \in \mathbb{R}^{m \times n}$ denotes all the linear coefficients.

---

**Algorithm 1** Algorithm for solving the LDD-L1-SC problem

---

**Initialize** $\mathbf{W}$ and $\mathbf{A}$
**repeat**
   Update $\mathbf{A}$ with $\mathbf{W}$ being fixed, by solving $n$ Lasso problems defined in Eq.(6).
   **repeat**
      Update $\widetilde{\mathbf{W}}$ by solving the Lasso problem defined in Eq.(10)
      $\mathbf{U} \leftarrow \mathbf{U} + (\mathbf{W} - \widetilde{\mathbf{W}})$
      **repeat**
         **for** $i \leftarrow 1$ to $n$ **do**
            Update the $i$th column vector $\mathbf{w}_i$ of $\mathbf{W}$ using Eq.(23)
         **end for**
      **until** convergence of the problem defined in Eq.(12)
   **until** convergence of the problem defined in Eq.(9)
**until** convergence of the problem defined in Eq.(4)

---

## 3 ALGORITHM

For LDD-L1-NNs, a simple subgradient descent algorithm is applied to learn the weight parameters. For LDD-L1-SC, we solve it by alternating between $\mathbf{A}$ and $\mathbf{W}$: (1) updating $\mathbf{A}$ with $\mathbf{W}$ fixed; (2) updating $\mathbf{W}$ with $\mathbf{A}$ fixed. These two steps alternate until convergence. With $\mathbf{W}$ fixed, the subproblem defined over $\mathbf{A}$ is

$$\min_{\mathbf{A}} \quad \tfrac{1}{2}\|\mathbf{X} - \mathbf{W}\mathbf{A}\|_F^2 + \lambda_1 |\mathbf{A}|_1 \tag{5}$$

which can be decomposed into $n$ Lasso problems: for $i = 1, \cdots, n$

$$\min_{\mathbf{a}_i} \quad \tfrac{1}{2}\|\mathbf{x}_i - \mathbf{W}\mathbf{a}_i\|_2^2 + \lambda_1 |\mathbf{a}_i|_1 \tag{6}$$

where $\mathbf{a}_i$ is the coefficient vector of the $i$-th sample. Lasso can be solved by many algorithms, such as proximal gradient descent (PGD) (Parikh & Boyd, 2014). Fixing $\mathbf{A}$, the sub-problem defined over $\mathbf{W}$ is:

$$\min_{\mathbf{W}} \quad \tfrac{1}{2}\|\mathbf{X} - \mathbf{W}\mathbf{A}\|_F^2 + \tfrac{\lambda_2}{2}\|\mathbf{W}\|_F^2 + \tfrac{\lambda_3}{2}(\operatorname{tr}(\mathbf{W}^\top \mathbf{W}) - \log\det(\mathbf{W}^\top \mathbf{W})) + \lambda_4 |\mathbf{W}|_1. \tag{7}$$

We solve this problem using an ADMM-based algorithm. First, we write the problem into an equivalent form

$$\begin{aligned} \min_{\mathbf{W}} \quad & \tfrac{1}{2}\|\mathbf{X} - \mathbf{W}\mathbf{A}\|_F^2 + \tfrac{\lambda_2}{2}\|\mathbf{W}\|_F^2 + \tfrac{\lambda_3}{2}(\operatorname{tr}(\mathbf{W}^\top \mathbf{W}) - \log\det(\mathbf{W}^\top \mathbf{W})) + \lambda_4 |\widetilde{\mathbf{W}}|_1 \\ s.t. \quad & \mathbf{W} = \widetilde{\mathbf{W}} \end{aligned} \tag{8}$$

Then we write down the augmented Lagrangian function

$$\begin{aligned} & \tfrac{1}{2}\|\mathbf{X} - \mathbf{W}\mathbf{A}\|_F^2 + \tfrac{\lambda_2}{2}\|\mathbf{W}\|_F^2 + \tfrac{\lambda_3}{2}(\operatorname{tr}(\mathbf{W}^\top \mathbf{W}) - \log\det(\mathbf{W}^\top \mathbf{W})) + \lambda_4 |\widetilde{\mathbf{W}}|_1 \\ & + \langle \mathbf{U}, \mathbf{W} - \widetilde{\mathbf{W}} \rangle + \tfrac{\rho}{2}\|\mathbf{W} - \widetilde{\mathbf{W}}\|_F^2. \end{aligned} \tag{9}$$

We minimize this Lagrangian function by alternating among $\widetilde{\mathbf{W}}$, $\mathbf{U}$ and $\mathbf{W}$.

**Update $\widetilde{\mathbf{W}}$** The subproblem defined on $\widetilde{\mathbf{W}}$ is

$$\min_{\widetilde{\mathbf{W}}} \quad \lambda_4 |\widetilde{\mathbf{W}}|_1 - \langle \mathbf{U}, \widetilde{\mathbf{W}} \rangle + \tfrac{\rho}{2}\|\mathbf{W} - \widetilde{\mathbf{W}}\|_F^2 \tag{10}$$

which is a Lasso problem and can be solved using PGD (Parikh & Boyd, 2014).

**Update $\mathbf{U}$** The update equation of $\mathbf{U}$ is simple.

$$\mathbf{U} = \mathbf{U} + (\mathbf{W} - \widetilde{\mathbf{W}}) \tag{11}$$

The subproblem defined on $\mathbf{W}$ is

$$\begin{aligned} \min_{\mathbf{W}} \quad & \tfrac{1}{2}\|\mathbf{X} - \mathbf{W}\mathbf{A}\|_F^2 + \tfrac{\lambda_2}{2}\|\mathbf{W}\|_F^2 + \tfrac{\lambda_3}{2}(\operatorname{tr}(\mathbf{W}^\top \mathbf{W}) - \log\det(\mathbf{W}^\top \mathbf{W})) + \langle \mathbf{U}, \mathbf{W} \rangle \\ & + \tfrac{\rho}{2}\|\mathbf{W} - \widehat{\mathbf{W}}\|_F^2 \end{aligned} \tag{12}$$

which can be solved using a coordinate descent algorithm. The derivation is given in the next subsection.

### 3.1 COORDINATE DESCENT ALGORITHM FOR LEARNING $\mathbf{W}$

In each iteration of the CD algorithm, one basis vector is chosen for update while the others are fixed. Without loss of generality, we assume it is $\mathbf{w}_1$. The sub-problem defined over $\mathbf{w}_1$ is

$$\min_{\mathbf{w}_1} \quad \frac{1}{2} \sum_{i=1}^{n} \|\mathbf{x}_i - \sum_{l=2}^{m} a_{il}\mathbf{w}_l - a_{i1}\mathbf{w}_1\|_2^2 + \frac{\lambda_2+\lambda_3}{2}\|\mathbf{w}_1\|_2^2 \\ -\frac{\lambda_3}{2}\mathrm{logdet}(\mathbf{W}^\top\mathbf{W}) + \mathbf{u}^\top\mathbf{w}_1 + \frac{\rho}{2}\|\mathbf{w}_1 - \widetilde{\mathbf{w}}_1\|_2^2 \tag{13}$$

To obtain the optimal solution, we take the derivative of the objective function and set it to zero. First, we discuss how to compute the derivative of $\mathrm{logdet}(\mathbf{W}^\top\mathbf{W})$ w.r.t $\mathbf{w}_1$. According to the chain rule, we have

$$\frac{\partial\mathrm{logdet}(\mathbf{W}^\top\mathbf{W})}{\partial\mathbf{w}_1} = 2\mathbf{W}(\mathbf{W}^\top\mathbf{W})_{:,1}^{-1} \tag{14}$$

where $(\mathbf{W}^\top\mathbf{W})_{:,1}^{-1}$ denotes the first column of $(\mathbf{W}^\top\mathbf{W})^{-1}$. Let $\mathbf{W}_{\neg 1} = [\mathbf{w}_2, \cdots, \mathbf{w}_m]$, then

$$\mathbf{W}^\top\mathbf{W} = \begin{bmatrix} \mathbf{w}_1^\top\mathbf{w}_1 & \mathbf{w}_1^\top\mathbf{W}_{\neg 1} \\ \mathbf{W}_{\neg 1}^\top\mathbf{w}_1 & \mathbf{W}_{\neg 1}^\top\mathbf{W}_{\neg 1} \end{bmatrix} \tag{15}$$

According to the inverse of block matrix

$$\begin{bmatrix} \mathbf{A} & \mathbf{B} \\ \mathbf{C} & \mathbf{D} \end{bmatrix}^{-1} = \begin{bmatrix} \widetilde{\mathbf{A}} & \widetilde{\mathbf{B}} \\ \widetilde{\mathbf{C}} & \widetilde{\mathbf{D}} \end{bmatrix} \tag{16}$$

where $\widetilde{\mathbf{A}} = (\mathbf{A} - \mathbf{B}\mathbf{D}^{-1}\mathbf{C})^{-1}, \widetilde{\mathbf{B}} = -(\mathbf{A} - \mathbf{B}\mathbf{D}^{-1}\mathbf{C})^{-1}\mathbf{B}\mathbf{D}^{-1}, \widetilde{\mathbf{C}} = -\mathbf{D}^{-1}\mathbf{C}(\mathbf{A} - \mathbf{B}\mathbf{D}^{-1}\mathbf{C})^{-1}$, $\widetilde{\mathbf{D}} = \mathbf{D}^{-1} + \mathbf{D}^{-1}\mathbf{C}(\mathbf{A} - \mathbf{B}\mathbf{D}^{-1}\mathbf{C})^{-1}\mathbf{B}\mathbf{D}^{-1}$, we have $(\mathbf{W}^\top\mathbf{W})_{:,1}^{-1}$ equals $[\mathbf{a} \quad \mathbf{b}^\top]^\top$ where

$$\mathbf{a} = (\mathbf{w}_1^\top\mathbf{w}_1 - \mathbf{w}_1^\top\mathbf{W}_{\neg 1}(\mathbf{W}_{\neg 1}^\top\mathbf{W}_{\neg 1})^{-1}\mathbf{W}_{\neg 1}^\top\mathbf{w}_1)^{-1} \tag{17}$$

$$\mathbf{b} = -(\mathbf{W}_{\neg 1}^\top\mathbf{W}_{\neg 1})^{-1}\mathbf{W}_{\neg 1}^\top\mathbf{w}_1\mathbf{a} \tag{18}$$

Then

$$\mathbf{W}(\mathbf{W}^\top\mathbf{W})_{:,1}^{-1} = [\mathbf{w}_1 \quad \mathbf{W}_{\neg 1}]\begin{bmatrix}\mathbf{a}\\\mathbf{b}\end{bmatrix} = \frac{\mathbf{M}\mathbf{w}_1}{\mathbf{w}_1^\top\mathbf{M}\mathbf{w}_1}. \tag{19}$$

where

$$\mathbf{M} = \mathbf{I} - \mathbf{W}_{\neg 1}(\mathbf{W}_{\neg 1}^\top\mathbf{W}_{\neg 1})^{-1}\mathbf{W}_{\neg 1}^\top. \tag{20}$$

To this end, we obtain the full gradient of the objective function in Eq.(13):

$$\sum_{i=1}^{n} a_{i1}(a_{i1}\mathbf{w}_1 + \sum_{l=2}^{m} a_{il}\mathbf{w}_l - \mathbf{x}_i) + (\lambda_2 + \lambda_3)\mathbf{w}_1 - \lambda_3\frac{\mathbf{M}\mathbf{w}_1}{\mathbf{w}_1^\top\mathbf{M}\mathbf{w}_1} + \rho(\mathbf{w}_1 - \widetilde{\mathbf{w}}_1) + \mathbf{u}. \tag{21}$$

Setting the gradient to zero, we get

$$((\sum_{i=1}^{n} a_{i1}^2 + \lambda_2 + \lambda_3 + \rho)\mathbf{I} - \lambda_3\mathbf{M}/(\mathbf{w}_1^\top\mathbf{M}\mathbf{w}_1))\mathbf{w}_1 = \sum_{i=1}^{n} a_{i1}(\mathbf{x}_i - \sum_{l=2}^{m} a_{il}\mathbf{w}_l) - \mathbf{u} + \rho\widetilde{\mathbf{w}}_1. \tag{22}$$

Let $\gamma = \mathbf{w}_1^\top\mathbf{M}\mathbf{w}_1$, $c = \sum_{i=1}^{n} a_{i1}^2 + \lambda_2 + \lambda_3 + \rho$, $\mathbf{b} = \sum_{i=1}^{n} a_{i1}(\mathbf{x}_i - \sum_{l=2}^{m} a_{il}\mathbf{w}_l) - \mathbf{u} + \rho\widetilde{\mathbf{w}}_j$, then $(c\mathbf{I} - \frac{\lambda_3}{\gamma}\mathbf{M})\mathbf{w}_1 = \mathbf{b}$ and $\mathbf{w}_1 = (c\mathbf{I} - \frac{\lambda_3}{\gamma}\mathbf{M})^{-1}\mathbf{b}$. Let $\mathbf{U}\mathbf{\Sigma}\mathbf{U}^\top$ be the eigen decomposition of $\mathbf{M}$, we have

$$\mathbf{w}_1 = \gamma\mathbf{U}(\gamma c\mathbf{I} - \lambda_3\mathbf{\Sigma})^{-1}\mathbf{U}^\top\mathbf{b}. \tag{23}$$

Then

$$\begin{aligned} &\mathbf{w}_1^\top\mathbf{M}\mathbf{w}_1 \\ &= \gamma^2\mathbf{b}^\top\mathbf{U}(\gamma c\mathbf{I} - \lambda_3\mathbf{\Sigma})^{-1}\mathbf{U}^\top\mathbf{U}\mathbf{\Sigma}\mathbf{U}^\top\mathbf{U}(\gamma c\mathbf{I} - \lambda_3\mathbf{\Sigma})^{-1}\mathbf{U}^\top\mathbf{b} \\ &= \gamma^2\mathbf{b}^\top\mathbf{U}(\gamma c\mathbf{I} - \lambda_3\mathbf{\Sigma})^{-1}\mathbf{\Sigma}(\gamma c\mathbf{I} - \lambda_3\mathbf{\Sigma})^{-1}\mathbf{U}^\top\mathbf{b} \\ &= \gamma^2\sum_{s=1}^{d}\frac{(\mathbf{U}^\top\mathbf{b})_s^2\Sigma_{ss}}{(rc - \lambda_3\Sigma_{ss})^2} = \gamma \end{aligned} \tag{24}$$

The matrix $\mathbf{A} = \mathbf{W}_{\neg 1}(\mathbf{W}_{\neg 1}^\top\mathbf{W}_{\neg 1})^{-1}\mathbf{W}_{\neg 1}^\top$ is idempotent, i.e., $\mathbf{A}\mathbf{A} = \mathbf{A}$, and its rank is $m - 1$. According to the property of idempotent matrix, the first $m - 1$ eigenvalues of $\mathbf{A}$ equal to one and

the rest equal to zero. Thereafter, the first $m - 1$ eigenvalues of $\mathbf{M} = \mathbf{I} - \mathbf{A}$ equal to zero and the rest equal to one. Based on this property, Eq.(24) can be simplified as

$$\gamma \sum_{s=m}^{d} \frac{(\mathbf{U}^\top \mathbf{b})_s^2}{(rc - \lambda_3)^2} = 1 \qquad (25)$$

After simplification, it is a quadratic function where $\gamma$ has a closed form solution. Then we plug the solution of $\gamma$ into Eq.(23) to get the solution of $\mathbf{w}_1$.

## 4 EXPERIMENTS

In these section, we present experimental results. The studies were performed on three models: sparse coding (SC) for document modeling, long short-term memory (LSTM) (Hochreiter & Schmidhuber, 1997) network for language modeling and convolutional neural network (CNN) (Krizhevsky et al., 2012) for image classification.

### 4.1 DATASETS

We used four datasets. The SC experiments were conducted on two text datasets: 20-Newsgroups[2] (20-News) and Reuters Corpus[3] Volume 1 (RCV1). The 20-News dataset contains newsgroup documents belonging to 20 categories, where 11314, 3766 and 3766 documents were used for training, validation and testing respectively. The original RCV1 dataset contains documents belonging to 103 categories. Following (Cai & He, 2012), we chose the largest 4 categories which contain 9625 documents, to carry out the study. The number of training, validation and testing documents are 5775, 1925, 1925 respectively. For both datasets, stopwords were removed and all words were changed into lower-case. Top 1000 words with the highest document frequency were selected to form the vocabulary. We used tf-idf to represent documents and the feature vector of each document is normalized to have unit L2 norm.

The LSTM experiments were conducted on the Penn Treebank (PTB) dataset (Marcus et al., 1993), which consists of 923K training, 73K validation, and 82K test words. Following (Mikolov et al.), top 10K words with highest frequency were selected to form the vocabulary. All other words are replaced with a special token UNK.

The CNN experiments were performed on the CIFAR-10 dataset[4]. It consists of 32x32 color images belonging to 10 categories, where 50,000 images were used for training and 10,000 for testing. 5000 training images were used as the validation set for hyperparameter tuning. We augmented the dataset by first zero-padding the images with 4 pixels on each side, then randomly cropping the padded images to reproduce 32x32 images.

### 4.2 LDD-L1 AND NON-OVERLAPNESS

First of all, we verify whether the LDD-L1 regularizer is able to promote non-overlapness. The study is performed on the SC model and the 20-News dataset. The number of basis vectors was set to 50. For 5 choices of the regularization parameter of LDD-L1: $\{10^{-4}, 10^{-3}, \cdots, 1\}$, we ran the LDD-L1-SC model until convergence and measured the overlap score (defined in Eq.2) of the basis vectors. The tradeoff parameter $\gamma$ inside LDD-L1 is set to 1. Figure 2 shows that the overlap score consistently decreases as the regularization parameter of LDD-L1 increases, which implies that LDD-L1 can effectively encourage non-overlapness. As a comparison, we replaced LDD-L1 with LDD-only and L1-only, and measured the overlap scores. As can be seen, for LDD-only, the overlap score remains to be 1 when the regularization parameter increases, which indicates that LDD alone is not able to reduce overlap. This is because under LDD-only, the vectors remain dense, which renders their supports to be completely overlapped. Under the same regularization parameter, LDD-L1 achieves lower overlap score than L1, which suggests that LDD-L1 is more effective in promoting non-overlapness. Given that $\gamma$ – the tradeoff parameter associated with the L1 norm in

---

[2] http://qwone.com/~jason/20Newsgroups/
[3] http://www.daviddlewis.com/resources/testcollections/rcv1/
[4] https://www.cs.toronto.edu/~kriz/cifar.html

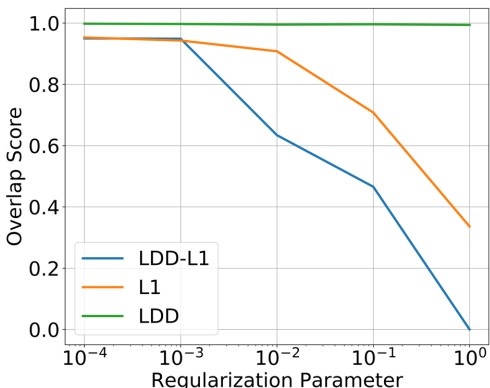

Figure 2: Overlap score versus the regularization parameter

| Vector | Representative Words |
|--------|----------------------|
| 1 | crime, guns |
| 2 | faith, trust |
| 3 | worked, manager |
| 4 | weapons, citizens |
| 5 | board, uiuc |
| 6 | application, performance, ideas |
| 7 | service, quality |
| 8 | bible, moral |
| 9 | christ, jews, land, faq |

Table 1: Representative words of 9 exemplar basis vectors

LDD-L1 – is set to 1, the same regularization parameter $\lambda$ imposes the same level of sparsity for both LDD-L1 and L1-only. Since LDD-L1 encourages the vectors to be mutually orthogonal, the intersection between vectors' supports is small, which consequently results in small overlap. This is not the case for L1-only, which hence is less effective in reducing overlap.

### 4.3 INTERPRETABILITY

In this section, we examine whether the weight vectors learned under LDD-L1 regularization are more interpretable, using SC as a study case. For each basis vector $\mathbf{w}$ learned by LDD-L1-SC on the 20-News dataset, we use the words (referred to as *representative words*) that correspond to the supports of $\mathbf{w}$ to interpret $\mathbf{w}$. Table 1 shows the representative words of 9 exemplar vectors. By analyzing the representative words, we can see vector 1-9 represent the following semantics respectively: crime, faith, job, war, university, research, service, religion and Jews. The representative words of these vectors have no overlap. As a result, it is easy to associate each vector with a unique concept, in other words, easy to interpret. Figure 3 visualizes the learned vectors where the black dots denote vectors' supports. As can be seen, the supports of different basis vectors are landed over different words and their overlap is very small.

### 4.4 REDUCING OVERFITTING

In this section, we verify whether LDD-L1 is able to reduce overfitting. The studies were performed on SC, LSTM and CNN. In each experiment, the hyperparameters were tuned on the validation set.

**Sparse Coding** For 20-News, the number of basis vectors in LDD-L1-SC is set to 50. $\lambda_1$, $\lambda_2$, $\lambda_3$ and $\lambda_4$ are set to 1, 1, 0.1 and 0.001 respectively. For RCV1, the number of basis vectors is set to 200. $\lambda_1$, $\lambda_2$, $\lambda_3$ and $\lambda_4$ are set to 0.01, 1, 1 and 1 respectively. We compared LDD-L1 with LDD-only and L1-only.

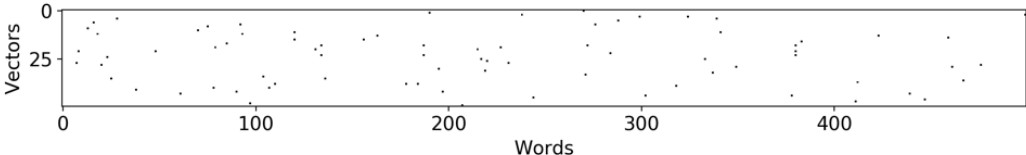

Figure 3: Visualization of basis vectors

| Method | 20-News | | RCV1 | |
|---|---|---|---|---|
| – | Test | Gap between train and test | Test | Gap between train and test |
| SC | 0.592 | 0.119 | 0.872 | 0.009 |
| LDD-SC | 0.605 | 0.108 | 0.886 | 0.005 |
| L1-SC | 0.606 | 0.105 | 0.897 | 0.005 |
| LDD-L1-SC | **0.612** | 0.099 | **0.909** | -0.015 |

Table 2: Classification accuracy on the test sets of 20-News and RCV1, and the gap between training and test accuracy.

To evaluate the model performance quantitatively, we applied the dictionary learned on the training data to infer the linear coefficients ($\mathbf{A}$ in Eq.4) of test documents, then performed $k$-nearest neighbors (KNN) classification on $\mathbf{A}$. Table 2 shows the classification accuracy on test sets of 20-News and RCV1 and the gap[5] between the accuracy on training and test sets. Without regularization, SC achieves a test accuracy of 0.592 on 20-News, which is lower than the training accuracy by 0.119. This suggests that an overfitting to training data occurs. With LDD-L1 regularization, the test accuracy is improved to 0.612 and the gap between training and test accuracy is reduced to 0.099, demonstrating the ability of LDD-L1 in alleviating overfitting. Though LDD alone and L1 alone improve test accuracy and reduce train/test gap, they perform less well than LDD-L1, which indicates that for overfitting reduction, encouraging non-overlapness is more effective than solely promoting orthogonality or solely promoting sparsity. Similar observations are made on the RCV1 dataset. Interestingly, the test accuracy achieved by LDD-L1-SC on RCV1 is even better than the training accuracy.

**LSTM for Language Modeling** The LSTM network architecture follows the word language model (PytorchTM) provided in Pytorch[6]. The number of hidden layers is set to 2. The embedding size is 1500. The size of hidden state is 1500. The word embedding and softmax weights are tied. The number of training epochs is 40. Dropout with 0.65 is used. The initial learning rate is 20. Gradient clipping threshold is 0.25. The size of mini-batch is 20. In LSTM training, the network is unrolled for 35 iterations. Perplexity is used for evaluating language modeling performance (lower is better). The weight parameters are initialized uniformly between [-0.1, 0.1]. The bias parameters are initialized as 0. We compare with the following regularizers: (1) L1 regularizer; (2) orthogonality-promoting regularizers based on cosine similarity (CS) (Yu et al., 2011), incoherence (IC) (Bao et al., 2013), mutual angle (MA) (Xie et al., 2015), decorrelation (DC) (Cogswell et al., 2015), angular constraint (AC) (Xie et al., 2017a) and LDD (Xie et al., 2017b).

Table 3 shows the perplexity on the PTB test set. Without regularization, PytorchLM achieves a perplexity of 72.3. With LDD-L1 regularization, the perplexity is significantly reduced to 71.1. This shows that LDD-L1 can effectively improve generalization performance. Compared with the sparsity-promoting L1 regularizer and orthogonality-promoting regularizers, LDD-L1 – which promotes non-overlapness by simultaneously promoting sparsity and orthogonality – achieves lower perplexity. For the convenience of readers, we also list the perplexity achieved by other state of the art deep learning models. The LDD-L1 regularizer can be applied to these models as well to potentially boost their performance.

---

[5]Training accuracy minus test accuracy.

[6]https://github.com/pytorch/examples/tree/master/word_language_model

| Network | Test |
|---|---|
| RNN (Mikolov & Zweig, 2012) | 124.7 |
| RNN+LDA (Mikolov & Zweig, 2012) | 113.7 |
| Deep RNN (Pascanu et al., 2013) | 107.5 |
| Sum-Product Network (Cheng et al., 2014) | 100.0 |
| RNN+LDA+KN-5+Cache (Mikolov & Zweig, 2012) | 92.0 |
| LSTM (medium) (Zaremba et al., 2014) | 82.7 |
| CharCNN (Kim et al., 2016) | 78.9 |
| LSTM (large) (Zaremba et al., 2014) | 78.4 |
| Variational LSTM with MC Dropout (Gal & Ghahramani, 2016) | 73.4 |
| PytorchLM | 72.3 |
| CS-PytorchLM (Yu et al., 2011) | 71.8 |
| IC-PytorchLM (Bao et al., 2013) | 71.9 |
| MA-PytorchLM (Xie et al., 2015) | 72.0 |
| DC-PytorchLM (Cogswell et al., 2015) | 72.2 |
| AC-PytorchLM (Xie et al., 2017a) | 71.5 |
| LDD-PytorchLM (Xie et al., 2017b) | 71.6 |
| L1-PytorchLM | 71.8 |
| LDD-L1-PytorchLM | 71.1 |
| Pointer Sentinel LSTM (Merity et al., 2016) | 70.9 |
| Ensemble of 38 Large LSTMs (Zaremba et al., 2014) | 68.7 |
| Ensemble of 10 Large Variational LSTMs (Gal & Ghahramani, 2016) | 68.7 |
| Variational RHN (Zilly et al., 2016) | 68.5 |
| Variational LSTM +REAL (Inan et al., 2016) | 68.5 |
| Neural Architecture Search (Zoph & Le, 2016) | 67.9 |
| Variational RHN +RE (Inan et al., 2016; Zilly et al., 2016) | 66.0 |
| Variational RHN + WT (Zilly et al., 2016) | 65.4 |
| Variational RHN + WT with MC dropout (Zilly et al., 2016) | 64.4 |
| Neural Architecture Search + WT V1 (Zoph & Le, 2016) | 64.0 |
| Neural Architecture Search + WT V2 (Zoph & Le, 2016) | 62.4 |

Table 3: Word-level perplexities on PTB test set

**CNN for Image Classification**   The CNN architecture follows that of the wide residual network (WideResNet) (Zagoruyko & Komodakis, 2016). The depth and width are set to 28 and 10 respectively. The networks are trained using SGD, where the epoch number is 200, the learning rate is set to 0.1 initially and is dropped by 0.2 at 60, 120 and 160 epochs, the minibatch size is 128 and the Nesterov momentum is 0.9. The dropout probability is 0.3 and the L2 weight decay is 0.0005. Model performance is measured using error rate, which is the median of 5 runs. We compared with (1) L1 regularizer; (2) orthogonality-promoting regularizers including CS, IC, MA, DC, AC, LDD and one based on locally constrained decorrelation (LCD) (Rodríguez et al., 2016).

Table 4 shows classification errors on CIFAR-10 test set. Compared with the unregularized WideResNet which achieves an error rate of 3.89%, the proposed LDD-L1 regularizer greatly reduces the error to 3.60%. LDD-L1 outperforms the L1 regularizer and orthogonality-promoting regularizers, demonstrating that encouraging non-overlapness is more effective than encouraging sparsity alone or orthogonality alone in reducing overfitting. The error rates achieved by other state of the art methods are also listed.

## 5   RELATED WORKS

The interpretation of representation learning models has been widely studied. Choi et al. (2016) develop a two-level neural attention model that detects influential variables in a reverse time order and use these variables to interpret predictions. Lipton (2016) discuss a taxonomy of both the desiderata and methods in interpretability research. Koh & Liang (2017) propose to use influence functions to trace a model's prediction back to its training data and identify training examples that are most relevant to a prediction. Dong et al. (2017) integrate topics extracted from human descriptions into neural networks via an interpretive loss and then use a prediction-difference maximization algo-

| Network | Error |
|---|---|
| Maxout (Goodfellow et al., 2013) | 9.38 |
| NiN (Lin et al., 2013) | 8.81 |
| DSN (Lee et al., 2015) | 7.97 |
| Highway Network (Srivastava et al., 2015) | 7.60 |
| All-CNN (Springenberg et al., 2014) | 7.25 |
| ResNet (He et al., 2016) | 6.61 |
| ELU-Network (Clevert et al., 2015) | 6.55 |
| LSUV (Mishkin & Matas, 2015) | 5.84 |
| Fract. Max-Pooling (Graham, 2014) | 4.50 |
| WideResNet (Huang et al., 2016) | 3.89 |
| CS-WideResNet (Yu et al., 2011) | 3.81 |
| IC-WideResNet (Bao et al., 2013) | 3.85 |
| MA-WideResNet (Xie et al., 2015) | 3.68 |
| DC-WideResNet (Cogswell et al., 2015) | 3.77 |
| LCD-WideResNet (Rodríguez et al., 2016) | 3.69 |
| AC-WideResNet (Xie et al., 2017a) | 3.63 |
| LDD-WideResNet (Xie et al., 2017b) | 3.65 |
| L1-WideResNet | 3.81 |
| LDD-L1-WideResNet | 3.60 |
| ResNeXt (Xie et al., 2016) | 3.58 |
| PyramidNet (Huang et al., 2016) | 3.48 |
| DenseNet (Huang et al., 2016) | 3.46 |
| PyramidSepDrop (Yamada et al., 2016) | 3.31 |

Table 4: Classification error (%) on CIFAR-10 test set

rithm to interpret the learned features of each neuron. Our method is orthogonal to these existing approaches and can be potentially used with them together to further improve interpretability.

## 6 Conclusions

In this paper, we propose a new type of regularization approach that encourages the weight vectors to have less-overlapped supports. The proposed LDD-L1 regularizer simultaneously encourages the weight vectors to be sparse and close to being orthogonal, which jointly produces the effects of less overlap. We apply this regularizer to two models: neural networks and sparse coding (SC), and derive an efficient ADMM-based algorithm for solving the regularized SC problem. Experiments on various datasets demonstrate the effectiveness of this regularizer in alleviating overfitting and improving interpretability.

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
