# OpenReview forum: "Learning Less-Overlapping Representations"
_ICLR.cc/2018/Conference — Reject_

### Official Review · AnonReviewer1 · 2017-11-22
**Novel orthogonality- and sparsity-promoting regularizer. Rather clear and technically sound paper, but incremental.**

**Rating:** 5
**Confidence:** 4

**Review:**

*Summary*
The paper introduces a matrix regularizer to simultaneously induce both sparsity and (approximate) orthogonality. The definition of the regularizer mostly relies on the previous proposal from Xie et al. 2017b, to which a weighted L1 term is added.
The regularizer aims at reducing overlap among the learned matrices, and it is applied to various neural networks and sparse coding (SC) settings.
Most of the challenges of the paper concentrate on the optimization side.
The evaluation of the paper is based on 3 experiments: SC (to illustrate the gain in interpretability and the reduction in overfitting), LTSM (for a NLP task over PTB) and CNN (for a computer vision task over CIFAR-10).

The paper is overall clear and fairly well structured, but it suffers from several flaws, as next discussed.

*Detailed comments*
(mostly in linear order)

-The proposed regularization scheme seems closely related to the approach taken in [Zass2007]; a detailed discussion and potential comparison should be provided. In particular, the approach of [Zass2007] would lead to an easier optimization.

-The sparse coding formulation has an extremely heavy parametrization (4 regularization parameters + the optimization parameter for ADMM + the number of columns of W). It seems to me that the resulting approach may not be very practical.

-Sparse coding: More references to previous work are needed, such as references related to alternating schemes and proximal optimization for SC (in Sec. 3); e.g., see [Mairal2010,Jenatton2011] and numerous references therein.

-I would suggest to move the derivations of Sec. 3.1 into an appendix not to break the flow of the readers. The derivations look sound.

-Due to the use of ADMM, I think that only W_tilde is sparse (due to the prox update (10)), but W may not be. This point should be discussed. Is a "manual" thresholding applied thereafter?

-For equation (25), I would precise that the columns of U have to be properly ordered to make sure we can only look at those from s=m...d.

-More details about the optimization in the case of the neural networks should be discussed.

-Could another splitting for ADMM, based on the logdet to reuse ideas from [Banerjee2008,Friedman2008], be possible?

-In table 2., are those 3-decimal statistics significant? Any idea of the variability of those numbers?

-Interpretability: The paper focuses on the gain in interpretability thanks to the regularizer (e.g., Table 1 and 3). But all the proposed settings (SC or neural networks) are such that the parameters are themselves subject to sources of variations, e.g., the initial conditions. How can we make strong conclusions while inspecting the parameters?

-In Figure 2., it seems to be that the final performance metric should also be overlaid. What appears as interesting to me is the the trade-off between overlap score and final performance metric.

*References*

[Banerjee2008] Banerjee, O.; El Ghaoui, L. & d'Aspremont null, A. Model selection through sparse maximum likelihood estimation for multivariate Gaussian or binary data Journal of Machine Learning Research, MIT Press, 2008, 9, 485-516

[Friedman2008] Friedman, J.; Hastie, T. & Tibshirani, R. Sparse inverse covariance estimation with the graphical lasso Biostatistics, 2008, 9, 432

[Jenatton2011] Jenatton, R.; Mairal, J.; Obozinski, G. & Bach, F. Proximal Methods for Hierarchical Sparse Coding Journal of Machine Learning Research, 2011, 12, 2297-2334

[Mairal2010] Mairal, J.; Bach, F.; Ponce, J. & Sapiro, G. Online learning for matrix factorization and sparse coding Journal of Machine Learning Research, 2010, 11, 19-60

[Zass2007] Zass, R. & Shashua, A. Nonnegative sparse PCA Advances in Neural Information Processing Systems, 2007

---

### Official Review · AnonReviewer3 · 2017-11-26
**Interesting idea but insufficient explanations and experimental results**

**Rating:** 4
**Confidence:** 4

**Review:**

The paper studies a regularization method to promote sparsity and reduce the overlap among the supports of the weight vectors in the learned representations. The motivation of using this regularization is to enhance the interpretability of the learned representation and avoid overfitting of complex models.

To reduce the overlap among the supports of the weight vectors, an existing method (Xie et al, 2017b) encouraging orthogonality is adopted to let the Gram matrix of the weight vectors to be close to the identity matrix (so that each weight vector is with unit norm and any pair of vectors are approximately orthogonal).

Neural network and sparse coding are considered as two case studies. The alternating algorithm for solving the regularized sparse coding formulation is common and less attracted. I think the major point is to see how much benefit that the regularization can afford for learning deep neural networks. To avoid overfitting, some off-the-shelf methods, e.g., dropout which can be viewed as a kind of regularization, are commonly used for deep neural networks. Are there any connections between the adopted regularization terms and the existing methods? Will these less overlapped parameters control the activation of different neurons? I think these are some straightforward questions while there are not much explanations on those aspects.

For training neural networks, a simple sub-gradient method is used because of the non-smoothness of the regularization terms. When training with large neural networks, will the sub-gradient method affect the efficiency a lot compared without using the regularizer? For example, in the image classification problem with ResNet.

It is better not to use dropout in the experiments (language modeling and image classification), because one of the motivation of using the proposed regularizer is to avoid overfitting while dropout does the same work and may affect the evaluation of effectiveness of the regularization.

---

### Official Review · AnonReviewer2 · 2017-11-27
**logdet for diversity is not novel**

**Rating:** 3
**Confidence:** 5

**Review:**

The paper proposed a new regularization approach that simultaneously encourages the weight vectors (W) to be sparse and orthogonal to each other. The argument is that the sparsity helps to eliminate the irrelevant feature vectors by making the corresponding weights zero. Nearly orthogonal sparse vectors will have zeros at different indexes and hence, encourages the weight vectors to have small overlap in terms of indices of nonzero entries (called support). Small overlap in support of weight vectors, aids interpretability as each weight vector is associated with a unique subset of feature vectors. For example, in the topic model, small overlap encourages, each topic to have unique set of representation words.

The proposed approach used L1 regularizer for enforcing sparsity in W. For enforcing orthogonality between different weight vectors (wi, wj), the log-determinant divergence (LDD) regularization term encourages the Gram Matrix G (Gij = wiTwj) to be close to an identity matrix I. The authors applied and tested the performance of proposed approach on Neural Network and Sparse Coding (SC) machine learning models. The authors validated the need for their proposed regularizer through experiments on 4 datasets (3 text and 1 images).

Major
* The novelty of the paper is not clear. Neither L1 no logdet() are novel regularizers (see the literature of Determinatal Point Process). With the presence of the auto-differentiator, one cannot claim the making derivative a novelty.

* L1 is also encourages diversity although as explicit as logdet. This is also obvious from Fig 2.  Perhaps the advantage of diversity is in interpretability but that is hard to quantify and the authors did not put enough effort to do that; we only have small anecdotal results in section 4.3.

* The Table 1 is not convincing because one can argue, for example, gun (vec 1) and weapon (vec 4) are colinear.

* In section 4.2, the authors experimented with SC on text dataset.  The overlap score decreases as the strength of regularization increases. The authors didn’t show the effect of increasing the regularization strength on the model accuracy and convergence time. This analysis is important to make sure, the decrease in overlap score is not coming at the expense of model accuracy and performance.

* In section 4.4, increase in test set accuracy and difference between test and train set accuracy is used to validate the claim, that the proposed regularizer helps reducing over fitting. In Table-2, , the test accuracy increases between SC and LDD-L1 SC while the train accuracy remains almost the same. Also, the authors didn’t do any cross validation to support their claim. The difference is numbers is too small to support the claim.

* In section on LSTM for Language Modeling, the perplexity score of LDD-L1 regularization on PytorchLM received perplexity score of 1.2 lower than without regularization. Although, the author mentions it as a significant reduction, the lowest perplexity score in Table 3 is significantly lower than this result. It’s not clear how 1.2 reduction in perplexity is significant and why the method should be preferred while much better models already exists.

* Results of the best perplexity model, Neural Architecture Search + WT V2, with proposed regularization would also help, validating the generalizability claims of the new approach.

* In CNN for Image Classification section, details of increase interpretability of the model, in terms of classification decision, is missing.

* In Table-4, the proposed LDD-L1 WideResNet is not the best results. Results of adding the proposed regularization, to the best know method (Pyramid Sep Drop) would be interesting.

* The proposed regularization claims to provide more interpretable representation and less overfit model. The given experiments are inadequate to validate the claims.

* A more extensive experimentation is required to validate the applicability of the method.

* In SC, aj are the linear coefficients or the coefficient vector of the j-th sample. If A ∈ Rm×n then aj ∈ Rm×1 and j ranges between [1,n] as in equation 6. The notation in section 2.2, Sparse Coding section is misleading as j ranges between [1,m].

* In Related works, the authors mention previous work done on interpreting the results of the machine learning models. Related works on enhancing interpretability and reducing overfitting by using regularization is missing.

---

### Decision · Program_Chairs · 2018-01-29
**ICLR 2018 Conference Acceptance Decision**

**Decision:**

Reject

**Comment:**

Each of the reviewers had a slightly different set of issues with this paper but here is an attempt at a summary:

PROS:
1. Paper is mostly clear and well structured.

CONS:
1. Lack of novelty
2. Unsupported claims
3. Questionable methodology (using dropout confounds the goal of the experiment)

The authors did not submit a rebuttal.